# The Attenuated Secretion of Hyaluronan by UVA-Exposed Human Fibroblasts Is Associated with Up- and Downregulation of HYBID and HAS2 Expression via Activated and Inactivated Signaling of the p38/ATF2 and JAK2/STAT3 Cascades

**DOI:** 10.3390/ijms22042057

**Published:** 2021-02-19

**Authors:** Shuko Terazawa, Mariko Takada, Yoriko Sato, Hiroaki Nakajima, Genji Imokawa

**Affiliations:** 1Center for Bioscience Research & Education, Utsunomiya University, Tochigi 321-8505, Japan; terazawa@cc.utsunomiya-u.ac.jp (S.T.); mrktkd0711@cc.utsunomiya-u.ac.jp (M.T.); yoriko@cc.utsunomiya-u.ac.jp (Y.S.); 2School of Bioscience and Biotechnology, Tokyo University of Technology, Tokyo 192-0982, Japan; ozzy_ozzy41@yahoo.co.jp

**Keywords:** ultraviolet A, hyaluronan, human dermal fibroblasts, hyaluronan synthases, HYaluronan Binding protein Involved in HA Depolymerization

## Abstract

Little is known about the effects on hyaluronan (HA) metabolism of UVA radiation. This study demonstrates that the secretion of HA by human dermal fibroblasts (HDFs) is downregulated by UVA, accompanied by the down- and upregulation of mRNA and protein levels of the HA-synthesizing enzyme (HAS2) and the HA-degrading protein, HYaluronan Binding protein Involved in HA Depolymerization(HYBID), respectively. Signaling analysis revealed that the exposure distinctly elicits activation of the p38/MSK1/CREB/c-Fos/AP-1 axis, the JNK/c-Jun axis, and the p38/ATF-2 axis, but downregulates the phosphorylation of NF-kB and JAK/STAT3. A signal inhibition study demonstrated that the inhibition of p38 significantly abrogates the UVA-accentuated mRNA level of HYBID. Furthermore, the inhibition of STAT3 significantly downregulates the level of HAS2 mRNA in non-UVA exposed HDFs. Analysis using siRNAs demonstrated that transfection of ATF-2 siRNA but not c-Fos siRNA abrogates the increased protein level of HYBID in UVA-exposed HDFs. An inhibitor of protein tyrosine phosphatase but not of protein serine/threonine phosphatase restored the diminished phosphorylation level of STAT3 at Tyr 705, accompanied by a significant abolishing effect on the decreased mRNA expression level of HAS2. Silencing with a protein tyrosine phosphatase PTP-Meg2 siRNA revealed that it abrogates the decreased phosphorylation of STAT3 at Tyr 705 in UVA-exposed HDFs. These findings suggest that the UVA-induced decrease in HA secretion by HDFs is attributable to the down- and upregulation of HAS2 and HYBID expression, respectively, changes that are mainly ascribed to the inactivated signaling of the STAT3 axis due to the activated tyrosine protein phosphatase PTP-Meg2 and the activated signaling of the p38/ATF2 axis, respectively.

## 1. Introduction

Hyaluronan (HA) is an essential constitutional biopolymer localized in the extracellular microenvironment of the dermis where it sustains the typical physicochemical and viscoelastic properties of the skin. The large polymer length and polyanionic charge of HA enables it to bind water predominantly, and contributes to volume expansion, turgidity, and elasticity of the skin, especially in the dermis [1], where the molecular weight of HA produced by dermal fibroblasts is much higher than that produced by epidermal keratinocytes. In frequently sun-exposed skin such as on the face, a marked deficiency of HA occurs in concert with its fragmentation to smaller molecules in the dermis [2,3]. The distinctly disrupted dermal changes due to sun exposure also include the fragmentation and loss of type I collagen fibrils, which have been ascribed to the enhanced expression of matrix-metalloprotease (MMP)-1 [4,5]. In addition, the three-dimensional linear microstructures of elastic fibers that are essential to support their elastic properties are abnormally curled and/or highly tortuous as a result of the accentuated expression of fibroblast-derived elastase neprilysin [1,6,7,8,9,10,11,12]. Such deleterious tissue damage in the dermis results in the decreased elasticity of the skin, which leads to the subsequently increased incidence of skin wrinkles and sagging [1]. In this connection, we have already demonstrated that the onset of skin wrinkle formation occurs following the decrease of cutaneous elastic properties as a prerequisite factor and that the loss of skin elasticity is significantly associated with both the clinical scores and the depth of wrinkles as evaluated by morphometric analysis [13,14]. This close relationship is also corroborated by our clinical observations that the formation or amelioration of facial wrinkles distinctly occurs accompanied by a marked loss or recovery of skin elastic properties, respectively [13,14,15].

As for the relationship between HA content and its molecular distribution in the dermis and cutaneous photoaging symptoms, Yoshida et al. [16,17] recently reported that a distinct decrease of HA with a shift toward a lower molecular weight occurs in the papillary dermis of photoaged facial skin, whose deficiency accompanies the degree of skin wrinkling and sagging observed at the same skin areas. Those studies strongly support the possibility that a deficiency and/or fragmentation of HA in photoaged skin is mechanistically associated at least in part with the formation of skin wrinkles and sagging.

HA is a linear biopolymer consisting of repeating disaccharides (d-glucuronic acid; 1,3-*N*-acetylglucosamine; 1,4-*N*-acetyl-glucosamine) that is synthesized from activated nucleotide sugars (UDP-glucuronic acid and UDP-*N*-acetylglucosamine) at the inner plasma membrane of various mammalian cells by HA synthases (HAS) that extrude into the extracellular space. There are three different isoforms of HAS (HAS1, HAS2, and HAS3) that reside in the plasma membrane [18]. Two different isoforms of hyaluronidase (HYAL) (HYAL1 and HYAL2) are reported to degrade HA into smaller discrete fragments [19] that can modulate inflammatory responses by activating Toll-like receptors 2 and 4 [20,21]. A third isoform of HYAL (HYAL3) also exists, but its properties are not well understood. The possible roles of HYALs in the depolymerization of HA by HDFs have been ruled out by demonstrating that a novel HA binding protein, recently designated as “HYBID”, serves as a key factor in the degradation of HA by normal HDFs [22]. Thus, it has been established that the balance of HAS and HYBID functions plays an essential role in modulating the level of HA secretion and its molecular profiles.

Among the various wavelengths of ultraviolet (UV) radiation, UVA but not UVB can penetrate to the dermis and directly elicit various cellular dysfunctions in dermal fibroblasts. Thus, in photoaged dermis and in UVA-exposed dermal fibroblasts, there is impaired synthesis or morphological alterations of matrix-proteins such as collagens and elastin that are secreted by dermal fibroblasts into the intercellular space of the dermis [8,23]. The expression and/or activation of many matrix-metalloproteinases are upregulated by UVA radiation in culture, which contribute to the fragmentation of matrix-proteins [8,23]. The synthesis of mucopolysacharrides (including HA) that are expressed at the plasma membrane also seem to be affected [16]. Therefore, several types of dermal cellular damage and abnormal loss or changes of matrix-proteins or mucopolysacharrides are implicated as causative factors for the formation of wrinkles and sagging in photoaged skin. It is likely that the direct effects of UVA on the dermis provide a greater impact than those caused by UVB via epithelial–mesenchymal paracrine cytokine interactions [10,11,12].

Although the effects of UVB radiation on HA content in the dermis and on its secretion by fibroblasts in culture are well understood [24,25,26], little is known about the effects of UVA on HA metabolism in the dermis. The deficiency of HA observed in the dermis of photoaged skin [16,17] prompted us to examine the effects of UVA on HA secretion by HDFs. The results of this study demonstrate that the secretion of HA by HDFs is downregulated by UVA (5–10 J/cm^2^) in a dose-dependent fashion. This UVA dose-dependent decrease in HA secretion was accompanied by the down- and upregulation of gene and protein expression levels of HA-synthesizing enzyme (HAS2) and the HA-degrading protein HYBID, respectively. Furthermore, we characterized the intracellular signaling mechanisms involved in the inhibitory and stimulatory effects of UVA on the expression of HAS2 and HYBID, respectively, in HDFs. The results show that the down- and upregulation of HAS2 and HYBID expression are mainly ascribed to the downregulated signaling of the JAK/STAT3 axis and the upregulated signaling of the p38/AP-1 axis, respectively, following UVA exposure.

## 2. Results

### 2.1. Effects of UVA on the Cytotoxic Properties of Human Dermal Fibroblasts (HDFs)

To determine whether UVA is cytotoxic to HDFs, we cultured HDFs for 72 h following UVA irradiation and their viability was evaluated by cellular morphology and by the 3-(4,5-di-methylthiazol-2-yl)-2,5-diphenyltetrazolium bromide (MTT) assay. The results showed that there was no morphological change in UVA-exposed HDFs and no significant decrease in cell viability of HDFs at 24, 48, and 72 h post-irradiation (Figure 1).

### 2.2. Effects of UVA on the Secretion of Hyaluronan (HA) by HDFs

We first examined the effects of UVA on the secretion of HA by HDFs that accumulated over 1, 48, or 72 h of culture. The level of HA secretion by HDFs was significantly decreased at 48 and at 72 h post-irradiation by UVA at doses of 5 or 10 J/cm^2^ compared with unirradiated controls (Figure 2).

### 2.3. Molecular Mass Distribution of Secreted HA

The molecular mass distribution of HA that accumulated in the conditioned medium at 72 h post-sham and UVA irradiation was examined by Shodex OHpak SB-807 HQ gel filtration chromatography. The molecular mass was widely distributed, averaging about 250 kDa overall (Figure 3). The average molecular mass and molecular mass distribution of HA in the irradiated HFs showed hardly any change at 72 h post-irradiation although the total amount of HA markedly decreased (Figure 3).

### 2.4. Effects of UVA on mRNA Expression Levels of HAS1/2/3 in HDFs

The diminished secretion of HA caused by UVA at 10 J/cm² was accompanied by the significantly downregulated expression level of HAS2 mRNA at 3 and 6 h (Figure 4a), while HAS1 mRNA expression was undetectable and the HAS3 mRNA level was not affected by UVA (Figure 4b). This distribution of the mRNA expression levels of HAS1/2/3 is consistent with a previous report [27].

### 2.5. Effects of UVA on mRNA Levels of Hyaluronidases (HYAL1/2/3) in HDFs

Exposure of HDFs to UVA at 10 J/cm^2^ did not affect the mRNA levels of HYAL1, HYAL2, or HYAL3 at 1, 3, 6, 12, or 24 h post-irradiation (Figure 5).

### 2.6. Effects of UVA on mRNA and Protein Levels of HYaluronan Binding Protein Involved in HA Depolymerization (HYBID) in HDFs

Exposure of HDFs to UVA significantly upregulated the mRNA level of HYBID at 6 and 12 h post-irradiation (Figure 6a) and significantly increased the protein level of HYBID at 24 h post-irradiation (Figure 6b).

### 2.7. Effects of UVA on Intracellular Signaling Pathways in HDFs

The phosphorylation of p38 and JNK (54 kDa) was definitely stimulated in HDFs at 5, 15, 30, and 60 min and at 30 and 60 min post-UVA irradiation, respectively (Figure 7), whereas ERK phosphorylation was unchanged compared with the unirradiated control during 5–60 min post-irradiation (Figure 7).

Downstream of p38, the phosphorylation of ATF2 (Figure 8a) and MSK1(90 kDa) (Thr581/Ser360) (Figure 8b) was markedly stimulated at 0~60 min and at 30~60 min post-irradiation, respectively, which suggests the marked activation of p38/ATF2 and p38/MSK1(Thr581/Ser360). Downstream of JNK, the UVA exposure markedly stimulated the phosphorylation of c-Jun (48 kDa) at 0–60 min post-irradiation (Figure 8b), which indicates the activation of the JNK/c-Jun axis.

Downstream of MSK1, the phosphorylation of CREB was markedly stimulated at 0–60 min post-irradiation (Figure 9), which indicates a marked activation of the p38/MSK1(Thr581/Ser360)/CREB axis.

On the other hand, the phosphorylation of NF-kB (Ser 276/536) was slightly decreased at 15–60 min post-irradiation (Figure 10).

Downstream of CREB, the protein expression of c-Fos was markedly increased at 60 min post-irradiation (Figure 11), which suggests the activation of the CREB/c-Fos/AP-1 axis.

Interestingly, the phosphorylation of STAT3 (Tyr705), which occurs downstream of JAK1/2, but not STAT3 (Ser727), which occurs downstream of ERK, was slightly decreased at 5–10 min and significantly downregulated at 15 min post-irradiation (Figure 12).

Consistently, the phosphorylation of JAK2(Tyr221), which occurs upstream of STAT3(Tyr705) was significantly downregulated at 5 min post-irradiation (Figure 13).

### 2.8. Effects of Signaling Inhibitors on mRNA Expression Levels of HAS2 and HYBID

As for activated signaling factors in UVA-exposed HDFs that may lead to the increased expression of HYBID mRNA, the addition of an inhibitor of p38, but not an inhibitor of JNK, significantly abrogated the increased mRNA level of HYBID at 12 h post-irradiation (Figure 14a,b). On the other hand, as for the inactivated signaling factors in UVA-exposed HDFs that may lead to the suppressed mRNA expression level of HAS2, the addition of an inhibitor of STAT3, but not an inhibitor of NF-kB, in unexposed HFs significantly downregulated the mRNA level of HAS2 at 3 h post-incubation (Figure 14c,d), similar to the decreased mRNA level of HAS2 observed in UVA-exposed HDFs at 3 h post-irradiation.

### 2.9. Effects of Transfection of c-Fos or ATF-2 siRNAs on the UVA-Induced Increase in HYBID Protein Level in HDFs

We next determined whether the UVA-stimulated level of HYBID protein would be interrupted by the silencing of ATF-2 and/or c-Fos. Whereas transfection of c-Fos siRNA did not affect the increased protein level of HYBID, even though accompanied by the decreased protein level of c-Fos (Figure 15a), the transfection of ATF-2 siRNA significantly abrogated the increased protein level of HYBID, accompanied by the significantly decreased protein level of ATF-2 (Figure 15b).

### 2.10. Effects of Inhibitors of Protein Phosphatases on the UVA-Induced Decrease in the Phosphorylation of STAT3 in HDFs

We next determined whether the UVA-diminished phosphorylation of STAT3 at Tyr703 is involved with an activation of protein tyrosine phosphatases in HDFs. When okadaic acid, an inhibitor of protein serine/threonine phosphatase, or sodium orthovanadate, an inhibitor of protein tyrosine phosphatase, was added to UVA-exposed HDFs, sodium orthovanadate at 80 µM, but not okadaic acid at 60 nM, distinctly abrogated the UVA-induced downregulation of the phosphorylation of STAT3 at Tyr 705 at 15 min post-irradiation (Figure 16a,b).

### 2.11. Effect of Sodium Orthovanadate on the UVA-Decreased mRNA Expression of HAS2 in HDFs

As for the signaling mechanisms involved in the UVA-decreased mRNA expression of HAS2 in HDFs, we determined whether sodium orthovanadate, an inhibitor of protein tyrosine phosphatase, abolishes the UVA-decreased mRNA expression of HAS2. While UVA significantly downregulated the gene expression level of HAS2, sodium orthovanadate significantly abrogated the UVA-suppressed mRNA expression level of HAS2 (Figure 17).

### 2.12. Effects of Transfection of a Protein Tyrosine Phosphatase (PTPMEG2) siRNA on the UVA-Induced Decrease in the Phosphorylation of STAT3 at Tyr 705 and JAK at Tyr 221 in HDFs

We next determined whether the UVA-suppressed phosphorylation of STAT3 at Tyr703 is interrupted by silencing the protein tyrosine phosphatase PTPMEG2, which is involved in the dephosphorylation of STAT3 at Tyr 705 [28]. The transfection of PTP-Meg2 siRNA significantly downregulated the mRNA levels of PTPMEG2 both in UVA- and in sham-irradiated HDFs at 3 h (Figure 18a). The same transfection with PTP-Meg2 siRNA distinctly abrogated the downregulated phosphorylation of STAT3 at Tyr 705 and of JAK2 at Tyr221, which was accompanied by the diminished protein level of PTP-Meg2 (Figure 18b,c). On the other hand, we were unable to determine the possible involvement of SHP1/2 [29] in UVA-inactivated signaling of JAK2/STAT3 because SHP1/2 mRNA was undetectable in HDFs.

## 3. Discussion

The results of this study demonstrate for the first time that exposure to UVA significantly diminishes the secretion of HA by HDFs in culture. Since a distinct deficiency and fragmentation of HA occurs in the dermis of sun-exposed skin [16], this result provides a deep insight into the photoaging mechanisms that underlie wrinkling and sagging of the skin. As levels of HA in extracellular circumstances are regulated by the HA synthetic enzymes HAS1/2/3 and by the HA degradative protein HYBID [22], analysis of mRNA levels of those factors using real time qPCR revealed that the inhibitory effect of UVA on HA secretion is distinctly accompanied by the up- and downregulation of mRNA levels of HYBID and HAS2, respectively. Consistently, western blotting analysis revealed that the protein level of HYBID was significantly upregulated by UVA. On the other hand, UVA significantly diminished the HA amounts in the conditioned medium without any changes in its molecular mass distribution, which indicates that the depolymerization process by HYBID does not contribute to the decreased level of HA secretion in UVA-exposed HFs in culture. It is likely that the downregulated expression of HAS2 is mainly responsible for the UVA-diminished secretion of HA in culture. Although this contrasts with the in vivo situation where a distinct deficiency and fragmentation of HA occurs in the dermis of sun-exposed skin [16], we thought it likely that the depolymerization process by HYBID in which secreted and membrane-associated HAs are integrated into cells and are fragmented by unknown degradative enzymes associated with HYBID [22] could not efficiently undergo in culture due to its diffusion into the medium, but definitely occurs in the UVA-exposed dermis. The sum of these findings indicates that under the in vivo skin conditions, the UVA irradiation penetrates to the dermis and causes HDFs to downregulate HAS2 expression and to simultaneously upregulate HYBID expression, which results in the distinct deficiency and fragmentation of HA in the dermis. Therefore, to develop signaling regulatory agents capable of preventing the UVA-induced deficiency and fragmentation of HA in vivo, it is important to identify signaling factors leading to the downregulated and the upregulated expression of HAS2 and HYBID, respectively.

To elucidate what signaling cascades are responsible for the downregulated and the upregulated gene expression of HAS2 and HYBID, respectively, we determined intracellular signaling pathways that are activated or inactivated by UVA with reference to the UVA- or UVB- induced activation or inactivation of intracellular signaling pathways in HDFs [30,31,32,33,34] and in human keratinocytes (HKs) [35,36,37]. In UVB-exposed HKs, sequential UVB-activated signaling lineages occur at the front lines of stress-activated signaling cascades including p38, ERK, and JNK in which they are significantly phosphorylated (activated) with a peak at 5–15 min post-irradiation by UVB [36]. The activation of these front lines of stress-activated signaling is followed by the subsequent activation of their downstream signaling cascades including NFкB, IKK, CK2/I-кBα, AP-1(c-Fos/c-Jun), ATF-2, and MSK1 [36]. On the other hand, our signaling analysis revealed that the exposure of HDFs to UVA similarly and significantly elicits activation of the p38/MSK1/CREB/c-Fos/AP-1 axis, the JNK/c-Jun axis, and the p38/ATF-2 axis, but downregulates the phosphorylation of NF-kB with ERK remaining unchanged. Our studies comparing UVB and UVA in terms of activated signaling cascades indicate that whereas the activated lineage of p38/NF-kB and p38/ATF2 predominantly occurs in UVB-exposed HKs, signaling cascades leading to AP-1 activation such as the p38/MSK1/CREB/c-Fos and the JNK/c-Jun axis are major lineages involved in UVA-activated signaling cascades in HDFs. Furthermore, the UVA-triggered signaling lineage is also characterized by an inactivation of the NF-kB axis, although the inactivation mechanism is unknown. Thus, in UVA-exposed HDFs, the activation of MSK1 by UVA does not contribute to the NF-kB activation via the phosphorylation of NF-kB at Ser 276, which was distinctly observed in UVB-exposed HKs [37]. Furthermore, its downstream signaling factor, activated CREB, is mainly associated with acting as a transcription factor with the upregulated protein level of c-Fos, which leads to the activation of its complex with c-Jun, namely AP-1 [36]. These results are consistent with several signaling mechanisms involved in the UVA-increased expression of MMP-1 [23,30,38] and of cathepsin L [39] by fibroblasts.

As for the signaling mechanisms involved in positively regulating HAS2 mRNA expression in HDFs, we have recently reported that mycosporine-like amino acids (MAAs), which are known to function as natural UV-absorbing compounds, stimulate the secretion of HA by upregulating HAS2 mRNA levels through activation of an intracellular signaling cascade consisting of p38/MSK1/CREB/c-Fos/AP-1 [40]. Since we thought it likely that activated and inactivated signaling cascades are mainly responsible for the up- and downregulated expression of HYBID and HAS2, respectively, the activation of the p38/MSK1/CREB/c-Fos/AP-1 axis, which contributes to the MAA-stimulated HAS2 mRNA expression [40] and also occurs in UVA-exposed HDFs, does not seem to be attributable to the downregulated gene expression of HAS2 in UVA-exposed HDFs. Thus, we characterized the effects of signaling inhibitors on mRNA levels of HAS2 and HYBID. Our signal inhibition study demonstrated that the inhibition of p38, but not of JNK, significantly abrogates the UVA-accentuated mRNA level of HYBID, which strongly suggests that activation of the p38/MSK1/CREB/c-Fos axis or the p38/ATF-2 axis, but not the JNK/c-Jun axis, is substantially associated with the UVA-stimulated expression of HYBID. On the other hand, our signal inhibition study revealed that the inhibition of NF-kB does not downregulate the level of HAS2 mRNA in unexposed HDFs, but rather significantly increases it to the same level, which strongly suggests that inactivation of the NFkB axis is not responsible for the downregulated expression of HAS2. In our signaling analysis of UVA-exposed HDFs, there were no inactivated signaling factors other than NF-kB. However, available evidence has demonstrated that the regulation of HAS2 expression is linked to several transcription factors including retinoic acid receptor [41], NF-kB [42], specificity protein 1 [43,44], CREB [45], and STAT3 [46]. Since CREB is activated in UVA-exposed HDFs, this allowed us to determine the effect of UVA on the STAT3 axis. Thus, it turned out that UVA significantly downregulates the phosphorylation of STAT3 in HDFs, which suggests its possible involvement in the UVA-suppressed mRNA expression of HAS2. As expected, our signal inhibition study revealed that the inhibition of STAT3 significantly downregulates the level of HAS2 mRNA in unexposed HDFs to a similar level as observed in UVA-exposed HDFs, which strongly suggests that inactivation of the STAT3 axis is mainly associated with the UVA-suppressed mRNA expression of HAS2.

Using transfection of c-Fos and ATF-2 siRNAs, we next determined which signaling pathways (i.e., the p38/MAPK-APK/MSK1/ CREB/c-Fos/AP-1 axis and/or the p38/ATF-2 axis) specifically contribute to the UVA-stimulated expression of HYBID. These silencing studies indicated that whereas the transfection of c-Fos siRNA does not abrogate the increased protein level of HYBID, the transfection of ATF-2 siRNA significantly abolishes the stimulated protein expression level of HYBID. This strongly suggests that the UVA-stimulated expression of HYBID is mediated via activation of the p38/ATF-2 axis, but not via the p38/MSK1/ CREB/c-Fos/AP-1 axis.

As for signaling mechanisms involved in the inactivation of the STAT3 axis, we thought it likely that protein phosphatases associated with these signaling factors may be activated by UVA, which would dephosphorylate and inactivate the corresponding phosphorylated signaling factors, leading to the downregulated expression of HAS2. Therefore, we determined whether inhibitors of protein phosphatases could abrogate the diminished phosphorylation level of STAT3 in UVA-exposed HDFs. It turned out that whereas an inhibitor of protein serine/threonine phosphatase, okadaic acid, does not exert a repairing effect on the decreased phosphorylation of STAT3 at Tyr 705 after UVA irradiation, an inhibitor of protein tyrosine phosphatase, sodium orthovanadate, slightly but distinctly abrogated the diminished phosphorylation level of STAT3 at Tyr 705, which was accompanied by a significant abolishing effect on the decreased mRNA expression of HAS2 in UVA-exposed HDFs. These findings strongly suggest that protein tyrosine phosphatases that are associated with STAT3 are activated by UVA, which results in the decreased phosphorylation of STAT3 at Tyr 705, leading to the downregulated mRNA expression of HAS2.

Available studies [47,48,49] attribute the phosphorylation of STAT3 to the activation of the JAK or Src kinase. JAK2-STAT3 signaling is also orchestrated by the balance between phosphorylation and dephosphorylation, the latter of which is mediated by several protein tyrosine phosphatases (PTPs), mainly including protein tyrosine phosphatase 1B (PTP1B) [50], SHP2 [29], TCPTP (PTPN2) [51], and PTPMeg2 (PTPN9, protein tyrosine phosphatase, nonreceptor type 9) both in the cytoplasm and in the nucleus [52]. Thus, protein tyrosine phosphatase 1B (PTP1B) modulates the cytokine signaling pathway by dephosphorylating JAK2 in the nucleus [50]. Other studies have demonstrated that STAT3 is dephosphorylated by SHP2 [29] and TCPTP (PTPN2) [51] in the nucleus, indicating a later phase of the dephosphorylation reaction. Recently, one study [28] reported that PTPMeg2 is a novel tyrosine phosphatase that directly interacts with STAT3 to mediate its dephosphorylation, especially in the cytoplasm, indicating an early phase of the dephosphorylation reaction. Overexpression or depletion of PTPMeg2 down- or upregulates, respectively, the phosphorylation of STAT3, which is paralleled by the attenuation of its transcription activity [28]. As TCPTP works predominantly in the nucleus with a delayed dephosphorylation reaction [51], our observation that the UVA-induced dephosphorylation of STAT3 occurs rapidly in the cytoplasm suggests that there is no involvement of TCPTP in the STAT3 inactivation. Furthermore, in the present study, SHP1/2 was undetectable at the mRNA level in HDFs. These findings prompted us to focus on PHP-Meg2 for its association with the inactivation of STAT3 in UVA-exposed HDFs. Silencing those protein tyrosine phosphatases revealed that transfection of PTP-Meg2 siRNA distinctly abrogates the decreased phosphorylation of STAT3 at Tyr 705 in UVA-exposed HDFs, in concert with the decreased gene and protein levels of PTP-Meg2. These findings indicate that UVA irradiation elicits the activation of protein tyrosine phosphatase PTP-Meg2, which results in the inactivation of STAT3, leading to the downregulated expression of HAS2 mRNA. On the other hand, we were unable to determine the effect of SHP1 siRNA on JAK2 phosphorylation because the silencing of SHP1 mRNA and reduced protein levels of SHP1 were not observed. Although the abrogating effect of PHP-Meg2 siRNA on the decreased phosphorylation of JAK2 at Tyr221 was beyond our expectation and requires further confirmation, these findings indicate that UVA irradiation elicits the activation of protein tyrosine phosphatase PTP-Meg2, which results in the inactivation of STAT3, leading to the downregulated expression of HAS2 mRNA. In conclusion, as depicted in Figure 19, the sum of these findings suggests that the UVA-induced decrease in HA secretion by HDFs is due to the down- and upregulation of HAS2 and HYBID expression, respectively. In turn, those changes are mainly ascribed to the downregulated signaling of the STAT3 axis by activated tyrosine protein phosphatases PTP-Meg2 and the upregulated signaling of the p38/ATF2 axis, respectively.

## 4. Materials and Methods

### 4.1. Materials

Antibodies to ERK, phospho-ERK, p38, phospho-p38, JNK, phospho-JNK, NFκBp65, phospho- Ser276/Ser536NFκBp65, CREB, phospho-Ser133CREB, c-Fos, ATF2, phospho-Thr71ATF2, MSK1 and phospho-Thr581/Ser 376/Ser360MSK1, JAK2 and phospho-Tyr221JAK2, STAT3 and phospho-Tyr705/Ser727STAT3 were purchased from Cell Signaling Technology (Danvers, MA, USA). Antibodies to β-actin and HYBID (Anti-KIAA1199) were obtained from Sigma-Aldrich Corp., St. Louis, MO, USA). The antibody to protein tyrosine phosphatase, nonreceptor type 9 (PTPMEG2/PTPN9) was purchased from Santa Cruz Biotechnology (Dallas, TX, USA). The NF-κB activation inhibitor II (JSH-23), the MEK inhibitor (U0126), and the STAT3 inhibitor were from Calbiochem (San Diego, CA, USA). The p38 inhibitor (SB239063) and the JNK inhibitor (JNK inhibitor II) were from Santa Cruz Biotechnology and Merck KGaA (Darmstadt, Germany), respectively. Okadaic acid and sodium orthovanadate were obtained from Calbiochem.

### 4.2. Cell Cultures

HDFs derived from human foreskins (Thermo Fisher Scientific, Waltham, MA, USA) were cultivated in Dulbecco’s modified Eagle’s medium (DMEM) with 10% fetal bovine serum (FBS) at 37 °C in a 95% air, 5% CO_2_ atmosphere.

### 4.3. Cell Viability Assay

Cell viability assays were performed using a Cell Proliferation Kit I (MTT assay) (Roche Diagnostics Corp., Indianapolis, IN, USA) according to the manufacturer’s instructions. HDFs were cultured for 72 h following UVA irradiation and their viability was evaluated using the MTT assay.

### 4.4. Measurement of HA

HDFs were seeded in DMEM with 10% FBS and were cultured for 12 h. After exchange with fresh DMEM without FBS, HA secreted into the culture medium was measured at the indicated times of culture using an HA Assay Kit (R&D Systems, Inc., Minneapolis, MN, USA), which can detect HA with molecular weight ≥35 kDa according to the manufacturer’s instructions. Levels of HA are expressed as ng/mL.

### 4.5. Real Time qRT-PCR

After exchange with fresh DMEM without FBS, levels of mRNAs encoding HAS1, HAS2, HAS3, HYBID, HYAL1, HYAL2, and HYAL3 in HDFs were measured at the indicated times of culture using real-time qRT-PCR. Total RNA was isolated from HDFs using a ReliaPrep™ RNA Miniprep System (Promega Corp., Madison, WI, USA), followed by reverse transcription to cDNA using a ReverTra Ace^®^ qPCR RT Master Mix (Toyobo Co. Ltd., Osaka, Japan). Real time PCR reactions were analyzed using TB Green Fast qPCR Mix (Takara Bio, Otsu, Shiga, Japan) and a Light Cycler 96(Roche Diagnostics, Mannheim, Germany). The primers used in this study are shown in Table 1. mRNA expression levels were corrected by the expression level of Glyceraldehyde-3-Phosphate Dehydrogenase (GAPDH) Ribosomal Protein Lateral Stalk Subunit P0 (RPLP0).

### 4.6. siRNA Transfection

Six × 104 HDFs were plated in each well of 12-well plates and were cultured for 1 day in DMEM with 5% FBS. On day 1, HDFs were transfected with a control siRNA (MISSION^®^ siRNA Universal Negative Control, Sigma Aldrich), a c-Fos siRNA (Mission siRNA, Sigma–Aldrich), an ATF2 siRNA (Santa Cruz), or an PTP-MEG2 siRNA (Santa Cruz) using Lipofectamine RNAiMAX Transfection Reagent (Invitrogen, Carlsbad, CA, USA) following the manufacturer’s protocol.

### 4.7. Western Blotting

After exchange with fresh DMEM without FBS, HDFs were cultured for the indicated times and were then solubilized in Radioimmunoprecipitation assay (RIPA) buffer (Wako, Osaka, Japan) plus protease and phosphatase inhibitors (Thermo Fisher Scientific) and were then centrifuged at 14,000 rpm at 4 °C for 20 min. The protein concentrations of the supernatants collected were determined using a Bicinchoninic Acid (BCA) Protein Assay Kit (Thermo Fisher Scientific). Proteins were adjusted to specific concentrations and were boiled at 95 °C for 5 min in 4X sample loading buffer (0.25 mol/l Tris-HCl, 8.0 *w/v*% SDS, 40.0 *w/v*% glycerol, 0.02 *w/v*% BPB, 20 vol% 2-mercaptoethanol, pH 6.8). Samples were separated on SDS-PAGE gels, transferred onto Polyvinylidene Difluoride (PVDF) membranes, blocked with 5% nonfat dry milk in Tris-buffered saline with Tween 20 (TBST), and probed with the primary antibodies noted above at 4 °C overnight. Membranes were washed three times for 15 min each with TBST, then probed with horseradish peroxidase-conjugated secondary antibodies (GE Healthcare Bioscience, Piscataway, NJ, USA) at room temperature for 1 h. They were then washed three times for 30 min each in TBST and specifically bound antibodies were detected using enhanced chemiluminescence (ECL, ECL prime, GE Healthcare Bioscience). Detection was performed using an imaging system (WSE-6100 LuminoGraph I; ATTO Corp., Tokyo, Japan) and a film processing machine (Max-Rhein, Asahiroentgen Ind. Co., Ltd., Kyoto, Japan).

### 4.8. UVA Irradiation Procedures

As the UVA irradiation source, 12 black-light lamps (FL20S·BLB, Panasonic, Osaka, Japan) were used and cells were situated under a glass plate to remove UVB included in the lamps. Lamp irradiance was measured using a UV Digital Radiometer (UV-340A, Lutron electronic, Taipei, Taiwan). The cells were rinsed in phosphate-buffered saline (PBS) and covered with a thin layer of Hanks’ balanced salt solution (HBSS) prior to UVA irradiation. Cells were exposed to UVA at doses of 5, 10, or 12 J/cm² in culture dishes without plate covers and mock-irradiated plates were wrapped with aluminum foil. As an example, UVA exposure at 10 J/cm² was comprised of UVA energy with 2.0 mw/cm² at 340 nm and 84 min of irradiation time. After the UVA irradiation, the HBSS was replaced with D-MEM without FBS at 37 °C.

### 4.9. Characterization of HA Molecular Mass Distribution

HA samples of the conditioned medium were concentrated by AmiconUltra-15(10K) (Merck KGaA, Darmstadt, Germany) and applied to a Shodex OHpak SB-807 HQ (Showa Denko, Tokyo, Japan) and eluted with 0.5 M NaCl. Fractions of 0.5 mL/min were collected and the HA concentration was measured by the HABP sandwich assays above-mentioned. HA with average molecular mass of 1515 kDa (NaHA-H2), 887 kDa (NaHA-M2), and 222 kDa (NaHA-L2) were obtained from PG Research (Tokyo, Japan) and used as standards.

### 4.10. Statistics

All data were expressed as means ± SD as indicated in the figure legends. For multiple comparisons, data were analyzed using the one-way ANOVA Tukey multiple comparison test or the two-way ANOVA Sidak’s multiple comparisons test. *p* values < 0.05 were considered statistically significant.

## Figures and Tables

**Figure 1 ijms-22-02057-f001:**
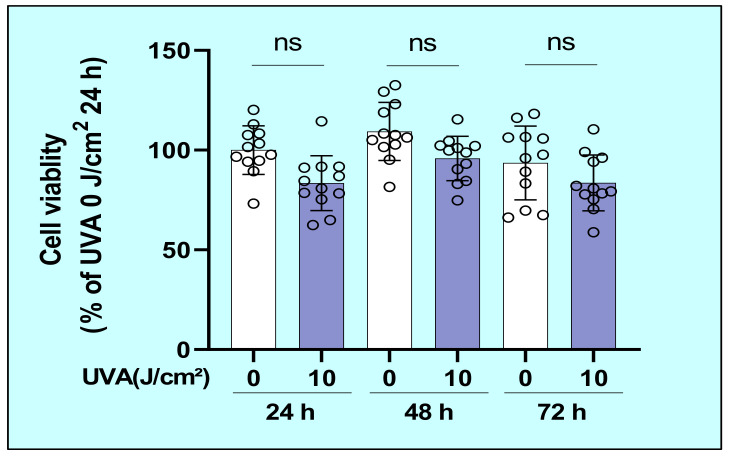
Effects of UVA on the viability of human dermal fibroblasts (HDFs) in culture. HDFs were cultured for 72 h following UVA irradiation at the indicated dose and their viability was evaluated using the MTT assay at the indicated time post-irradiation. Data represent means ± SD. *n* = 12.

**Figure 2 ijms-22-02057-f002:**
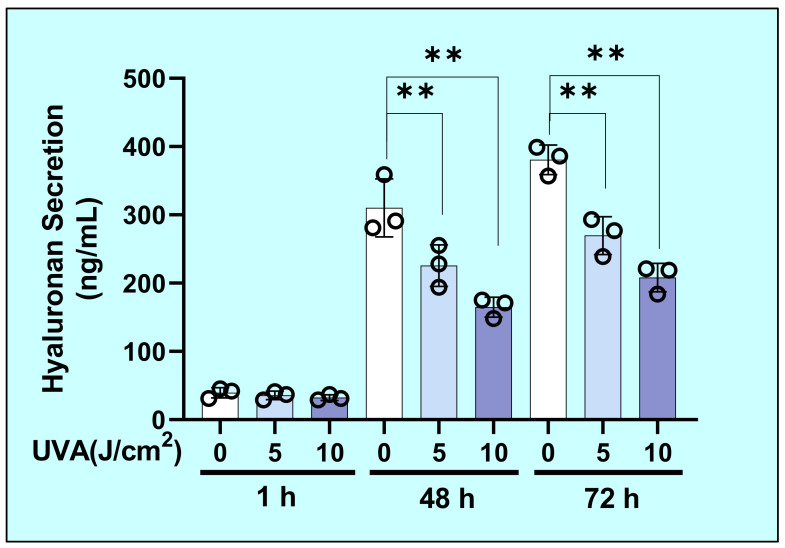
UVA decreases hyaluronan (HA) secretion by HDFs in culture. After HDFs were exposed to UVA at 0, 5, or 10 J/cm^2^, HA secreted into the culture medium was measured 1, 48, and 72 h later using an HA assay kit. Values are means ± S.D. from three independent. **: *p* < 0.01 vs. 0 J/cm^2^.

**Figure 3 ijms-22-02057-f003:**
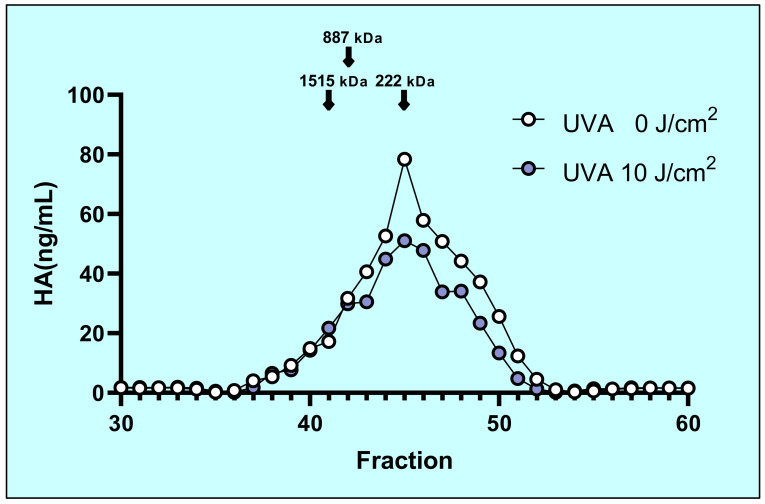
Changes of the molecular mass of HA after UVA irradiation. HA was extracted from the conditioned medium in non- (white circle) or UVA-irradiated (black circle) HFs at 48 h post-irradiation and applied to a Shodex OHpak SB-807 HQ column, and the HA concentration was measured by the Hyaluronan Binding Protein (HABP). sandwich assay. Each graph compares the non-UVA-irradiated molecular mass distribution with UVA-irradiated molecular mass distribution.

**Figure 4 ijms-22-02057-f004:**
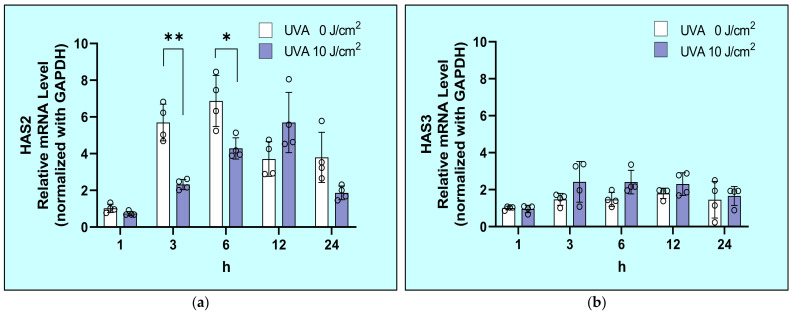
UVA specifically downregulates the transcription of HAS2 (**a**) but not HAS3 (**b**) in HDFs. After exposure to UVA at 0 or 10 J/cm^2^, HDFs were cultured in fetal bovine serum (DMEM) without fetal bovine serum (FBS). mRNA levels at the indicated times of culture were analyzed by Reverse transcription-quantitative polymerase chain reaction(RT-qPCR) as described in the Materials and Methods section. Values are means ± S.D. from four independent experiments. **: *p* < 0.01 vs. 0 J/cm^2^, *: *p* < 0.05 vs. 0 J/cm^2^.

**Figure 5 ijms-22-02057-f005:**
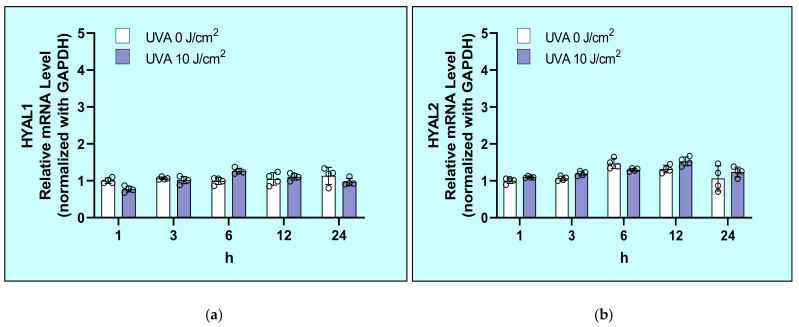
Effects of UVA on mRNA levels of hyaluronidases (HYAL1 (**a**) /2 (**b**) /3 (**c**)) in HDFs. After exposure to UVA at 0 or 10 J/cm^2^, HDFs were cultured in DMEM without FBS. mRNA levels of HYAL1/2/3 at the indicated times of culture were analyzed by real–time RT-PCR as described in the Materials and Methods section. Values are means ± S.D. from four independent experiments.

**Figure 6 ijms-22-02057-f006:**
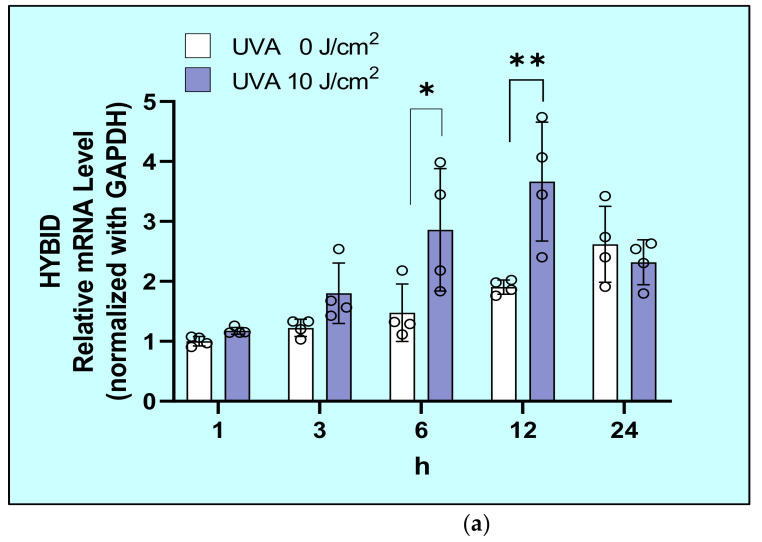
Effects of UVA on mRNA (**a**) and protein (**b**) levels of Hyaluronan Binding protein Involved in HA Depolymerization (HYBID). After exposure to UVA at 0, 10, or 12 J/cm^2^, HDFs were cultured in DMEM without FBS. Cell lysates obtained at the indicated times of culture were subjected to RT-qPCR and western blotting analysis as described in the Materials and Methods section. Values are means ± S.D. from four (**a**) and three (**b**) independent experiments. **: *p* < 0.01 vs. 0 J/cm^2^, *: *p* < 0.05 vs. 0 J/cm^2^.

**Figure 7 ijms-22-02057-f007:**
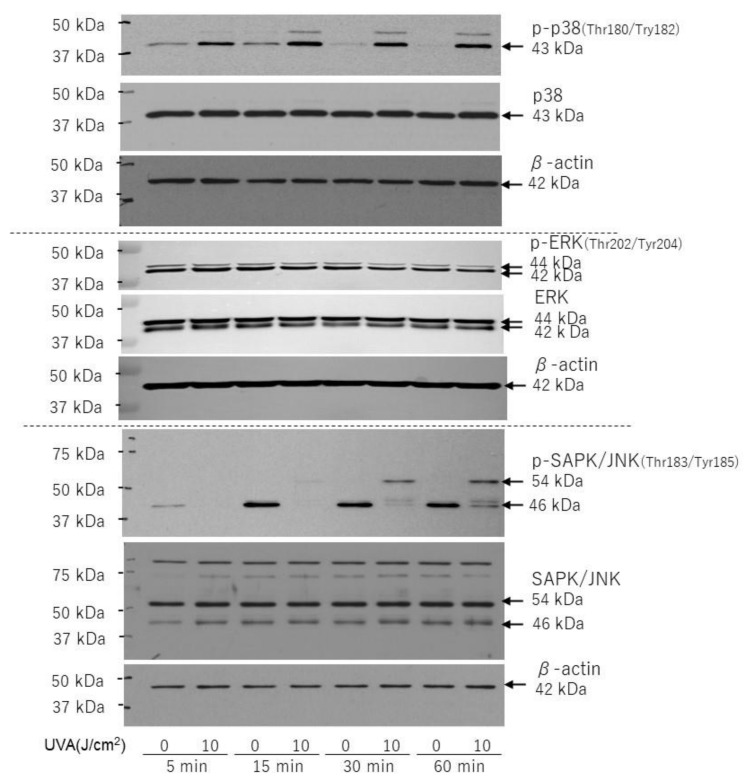
Effects of UVA on phosphorylation and protein levels of p38/ERK/JNK. HDFs were exposed to UVA at the indicated doses in culture and cell lysates prepared at the indicated times post-irradiation were subjected to western blotting as described in the Materials and Methods section. Representative immunoblots from three independent experiments are shown.

**Figure 8 ijms-22-02057-f008:**
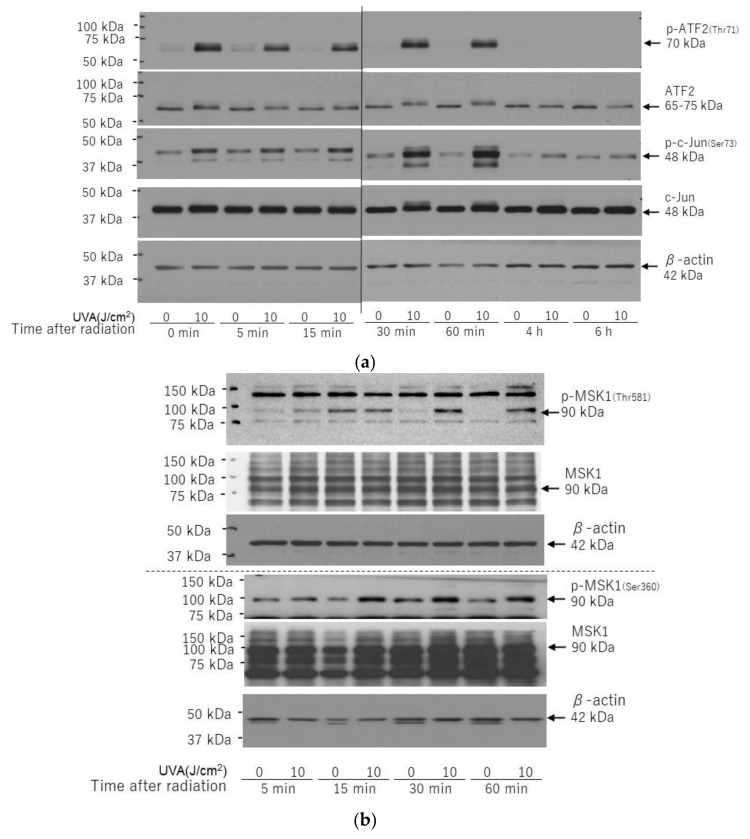
Effects of UVA on phosphorylation and protein levels of ATF2/c-Jun (**a**) and MSK1(Thr581/Ser360) (**b**). HDFs were exposed to UVA at the indicated doses in culture and cell lysates prepared at the indicated times post-irradiation were subjected to western blotting as described in the Materials and Methods section. Representative immunoblots from three independent experiments are shown.

**Figure 9 ijms-22-02057-f009:**
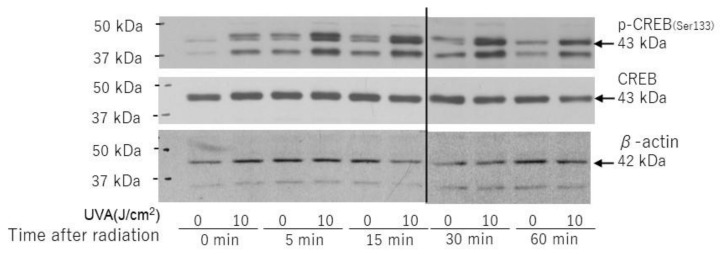
Effects of UVA on phosphorylation and protein levels of CREB. HDFs were exposed to UVA at the indicated doses in culture and cell lysates prepared at the indicated times post-irradiation were subjected to western blotting as described in the Materials and Methods section. Representative immunoblots from three independent experiments are shown.

**Figure 10 ijms-22-02057-f010:**
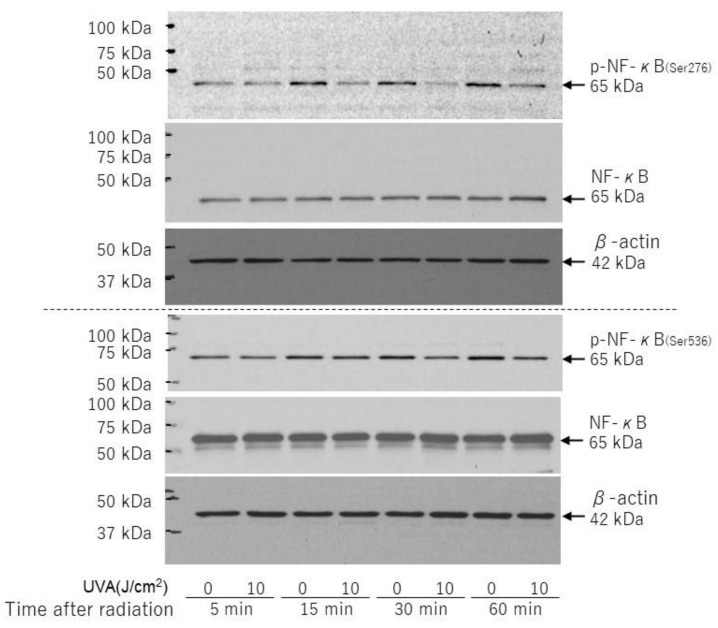
Effects of UVA on phosphorylation and protein levels of NF-kB(Ser276/Ser536). HDFs were exposed to UVA at the indicated doses in culture and cell lysates prepared at the indicated times post-irradiation were subjected to western blotting as described in the Materials and Methods section. Representative immunoblots from three independent experiments are shown.

**Figure 11 ijms-22-02057-f011:**
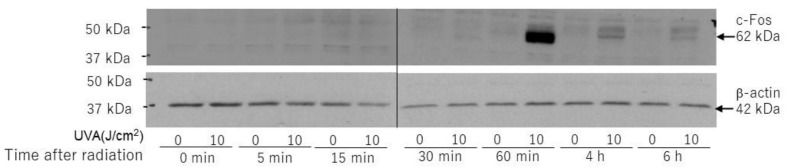
Effects of UVA on protein levels of c-Fos. HDFs were exposed to UVA at the indicated doses in culture and cell lysates prepared at the indicated times post-irradiation were subjected to western blotting as described in the Materials and Methods section. Representative immunoblots from three independent experiments are shown.

**Figure 12 ijms-22-02057-f012:**
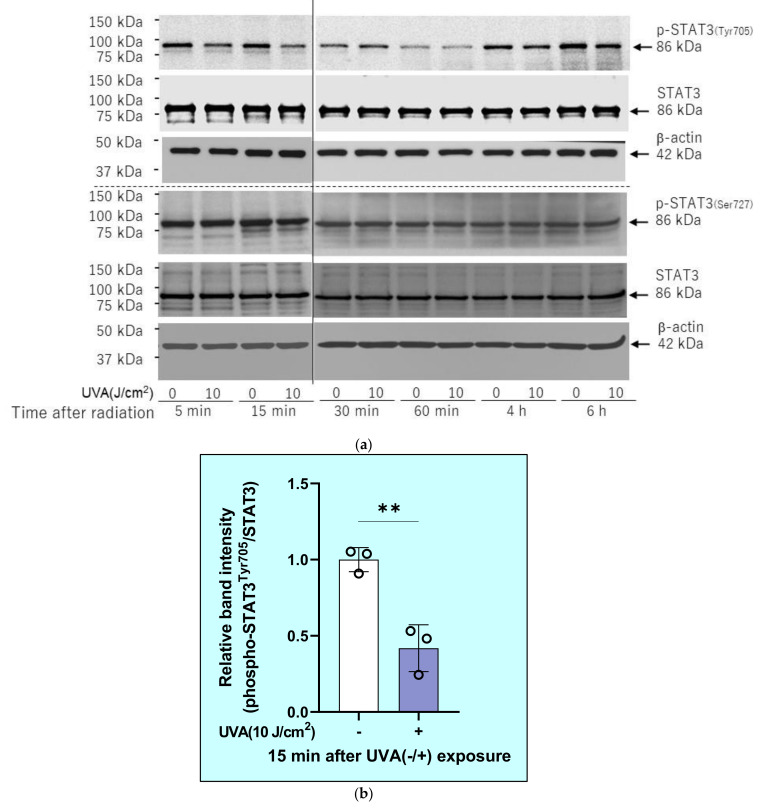
Effects of UVA on phosphorylation and protein levels of STAT3(Tyr705/Ser727). HDFs were exposed to UVA at the indicated doses in culture and cell lysates prepared at the indicated times post-irradiation were subjected to western blotting as described in the Materials and Methods section. Representative immunoblots from three independent experiments are shown. (**a**) Western blotting images. (**b**) Relative band intensity. Bands were quantitated and data represent means ± SD, *n* = 3, unpaired *t*-test **: *p* < 0.01 vs. 0 J/cm^2^.

**Figure 13 ijms-22-02057-f013:**
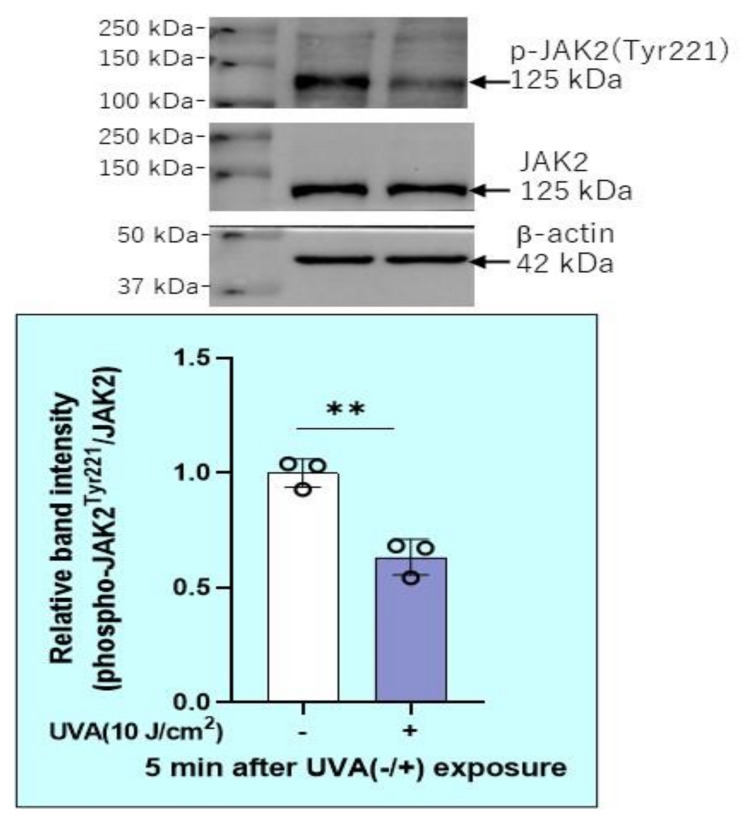
Effects of UVA on phosphorylation and protein levels of JAK2(Tyr221). HDFs were exposed to UVA at the indicated doses in culture and cell lysates prepared at the indicated times post-irradiation were subjected to western blotting as described in the Materials and Methods section. Representative immunoblots from three independent experiments are shown. Bands were quantitated and data represent means ± SD, *n* = 3, unpaired *t*-test **: *p* < 0.01 vs. 0 J/cm^2^.

**Figure 14 ijms-22-02057-f014:**
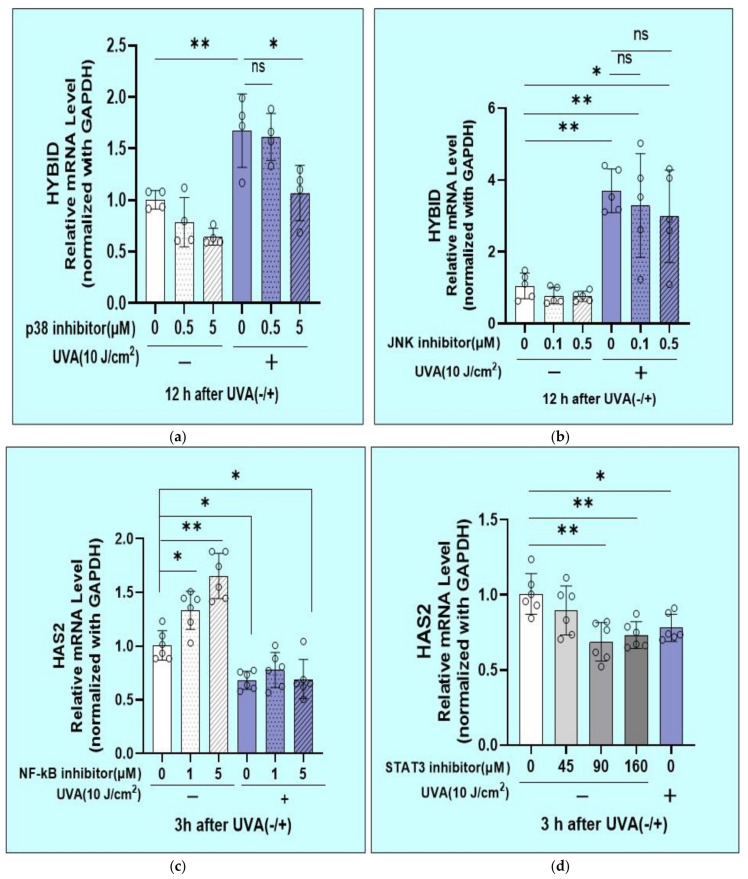
Effects of signaling inhibitors on mRNA expression levels of HAS2 and HYBID in HDFs. Effects of inhibitors of p38, JNK, NF-kB, and STAT3 on UVA- or sham-stimulated mRNA expression of HAS2 in HDFs. Signaling inhibitors were added at the indicated concentrations immediately after UVA irradiation at 10 J/cm^2^ or sham irradiation. Cell lysates prepared at the indicated times post-irradiation were subjected to RT-qPCR analysis. (**a**) p38 inhibitor, 12 h, *n* = 4, (**b**) JNK inhibitor, 12 h, *n* = 4, (**c**) NF-kB inhibitor, 3 h, *n* = 4, (**d**) STAT3 inhibitor, 3 h, *n* = 4, *: *p* < 0.05, **: *p* < 0.01.

**Figure 15 ijms-22-02057-f015:**
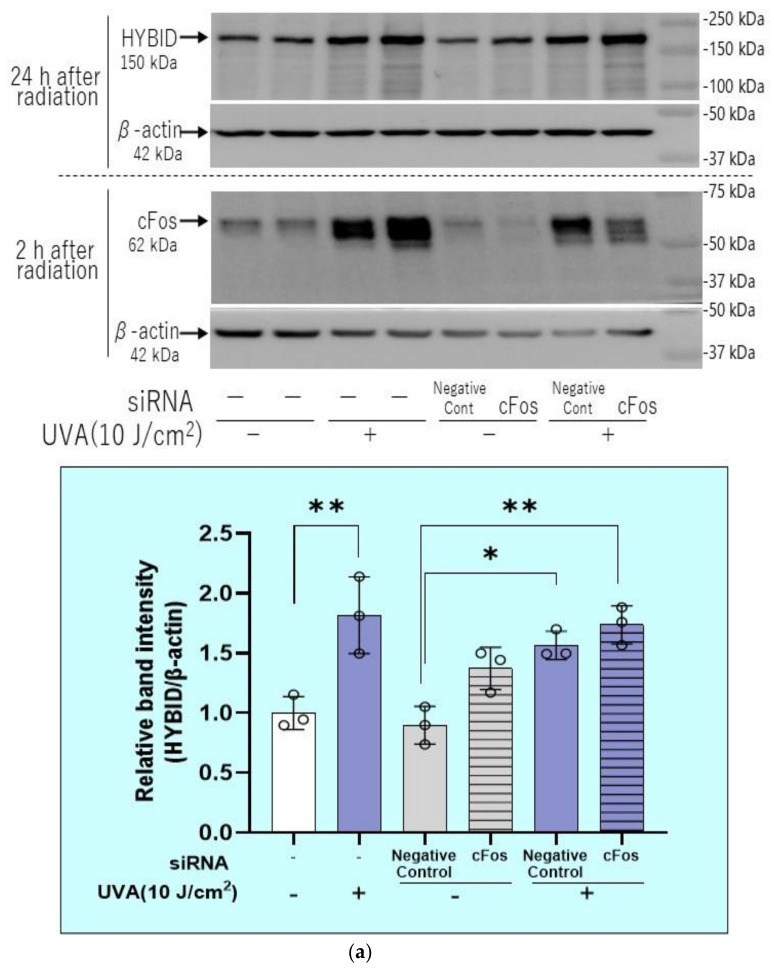
Effects of transfection of c-Fos (**a**) or ATF-2 (**b**) siRNA on the UVA-induced increase in the HYBID protein level in HDFs. (**a**) Effect of transfecting a c-Fos siRNA on the expression of c-Fos and HYBID proteins. (**b**) Effect of transfecting an ATF-2 siRNA on the expression of ATF-2 and HYBID proteins. HDFs were exposed to UVA at the indicated dose in culture and cell lysates prepared at the indicated times post-irradiation were subjected to western blotting as described in the Materials and Methods section. Representative immunoblots from three independent experiments are shown. Data represent means ± SD., *n* = 3, **: *p* < 0.01, *: *p* < 0.05.

**Figure 16 ijms-22-02057-f016:**
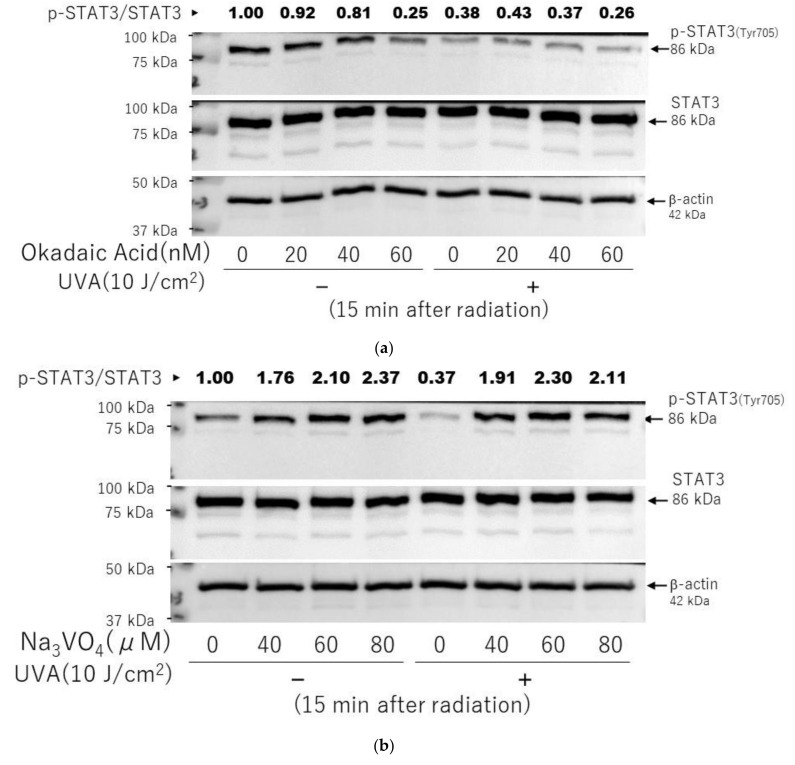
Effects of okadaic acid and sodium orthovanadate on the UVA-induced decrease in the phosphorylation of STAT3(Tyr705) in HDFs. (**a**) Effects of okadaic acid on the phosphorylation of STAT3(Tyr705). (**b**) Effects of sodium orthovanadate on the phosphorylation of STAT3(Tyr705). Sodium orthovanadate or okadaic acid was added at the indicated concentrations 2 h before and immediately after UVA (10 J/cm^2^) or sham irradiation. Cell lysates prepared at 15 min post-UVA or sham irradiation were subjected to western blotting as described in the Materials and Methods section. Representative immunoblots from three independent experiments are shown.

**Figure 17 ijms-22-02057-f017:**
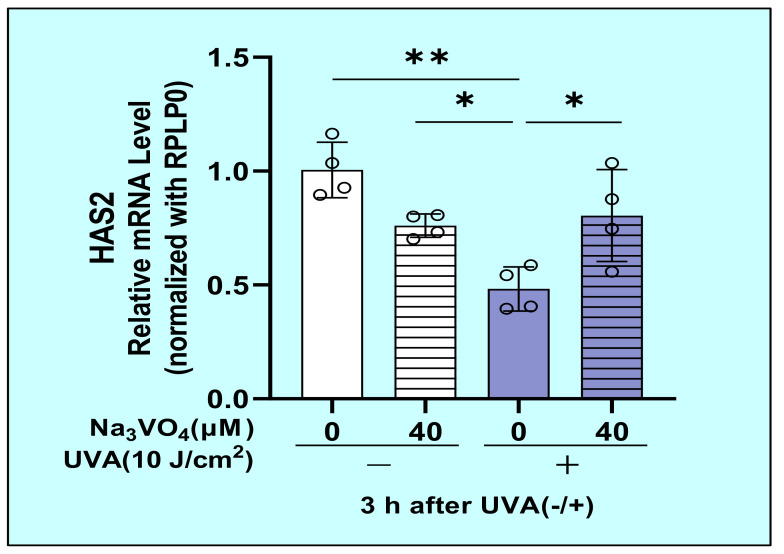
Effect of sodium orthovanadate on the UVA-decreased mRNA expression of HAS2 in HDFs. Sodium orthovanadate was added at 0 or 40 µM 2 h before and immediately after UVA (10 J/cm^2^) or sham irradiation. Cell lysate prepared at 3 h post-irradiation was subjected to RT-qPCR analysis as described in the Materials and Methods section. Data represent means ± SD., *n* = 4, ** *p* < 0.01, * *p* < 0.05.

**Figure 18 ijms-22-02057-f018:**
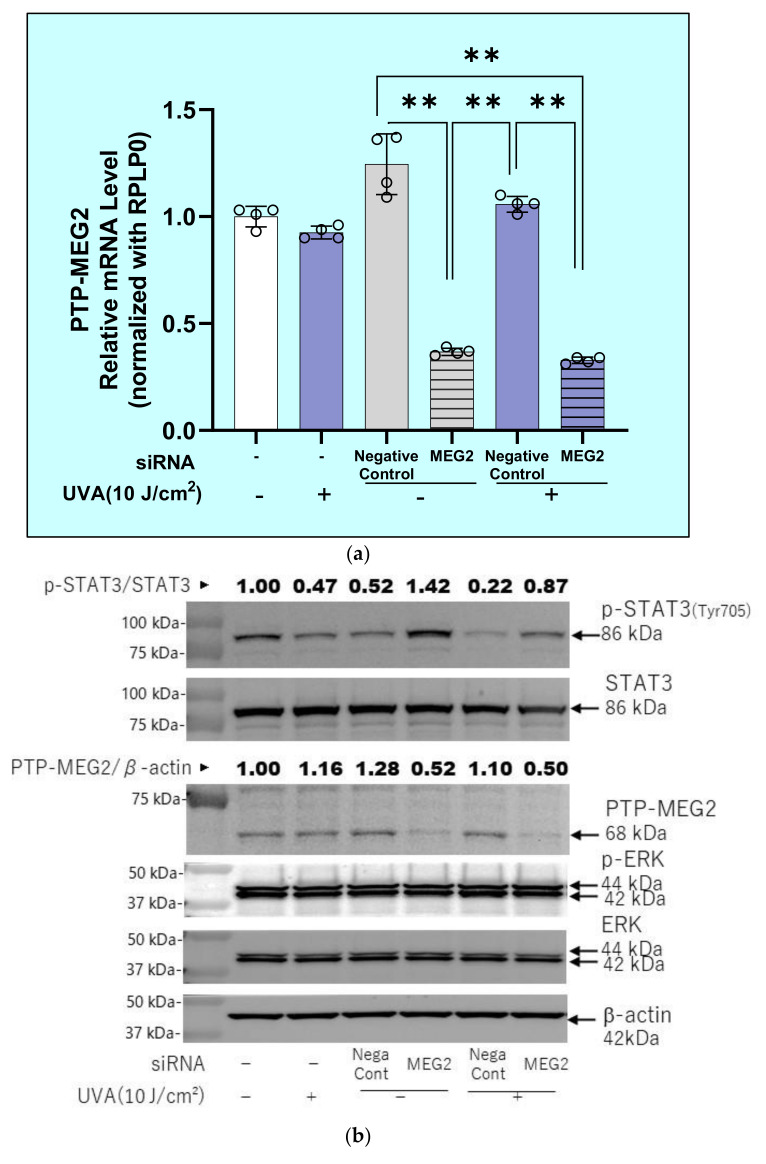
Effects of the transfection of a protein tyrosine phosphatase (PTPMEG2) siRNA on the UVA-induced decrease in the phosphorylation of STAT3 at Tyr 705 and JAK2 at Tyr 221 in HDFs. (**a**) Effect of transfecting a PTPMEG2 siRNA on the expression level of PTPMEG2 mRNA in UVA- or non-exposed HDFs at 3 h post-irradiation. Cell lysates were prepared at 3 h post-irradiation and were subjected to RT-qPCR analysis as described in the Materials and Methods section. Data represent means ± SD., *n* = 3, ** *p* < 0.01. (**b**) Effect of transfecting a PTPMEG2 siRNA on the expression level of PTPMEG2 protein and the phosphorylation level of STAT3 (Tyr705) and ERK in UVA- or non-exposed HDFs at 15 min post-irradiation. (**c**) Effect of transfecting a PTPMEG2 siRNA on the expression level of PTPMEG2 protein and the phosphorylation level of JAK2 (Tyr221) in UVA- or non-exposed HDFs at 5 min post-irradiation. Representative immunoblots from three independent experiments are shown.

**Figure 19 ijms-22-02057-f019:**
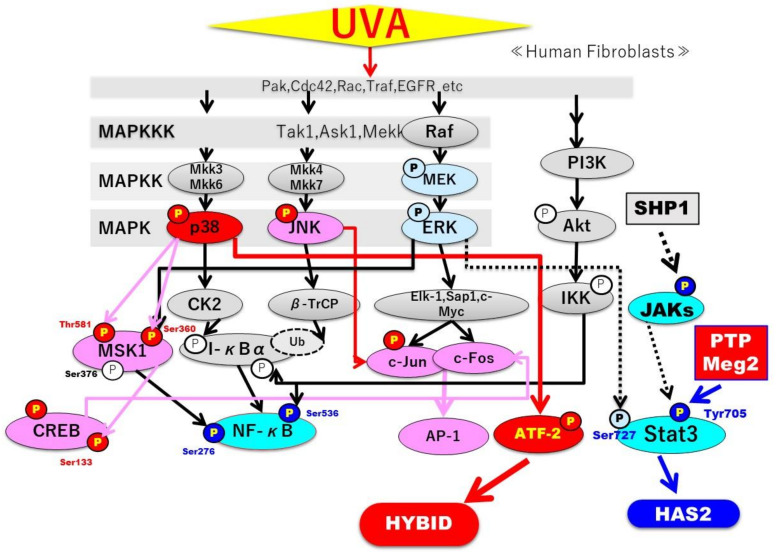
Activated and inactivated signaling cascades leading to the up-regulation and the down-regulation of the expression of HYBID and HAS2 in UVA-exposed HDFs. Factors that are activated by UVA are colored red/pink, while factors inactivated by UVA are colored light blue [36,40].

**Table 1 ijms-22-02057-t001:** Primers used in this study.

Primer Name	Sequence	
GAPDH	Forward	5’-GCACCGTCAAGGCGAGAAC-3’
	Reverse	5’-TGGTGAAGACGCCAGTGGA-3’
KIAA1199 (HYBID)	Forward	5’-CCAGGAATGTTGAATGTCT-3’
	Reverse	5’-ATTGGCTCTTGGTGAATG-3’
HAS1	Forward	5’-ATCCTGCATCAGCGGTCCTC-3’
	Reverse	5’-CTGGTTGTACCAGGCCTCAAGAA-3’
HAS2	Forward	5’-TCGCAACACGTAACGCAAT-3’
	Reverse	5’-ACTTCTCTTTTTCCACCCCATTT-3’
HAS3	Forward	5’-AACAAGTACGACTCATGGATTTCCT-3’
	Reverse	5’-GCCCGCTCCACGTTGA-3’
PTP-MEG2	Forward	5’-TTTATGCAGCTTGGCCCTGTC-3’
	Reverse	5’-ACACAGAACCCACCCAGTTTGAG-3’
HYAL1	Forward	5’-CGATATGGCCCAAGGCTTTAG-3
	Reverse	5’-ACCACATCGAAGACACTGACAT-3’
HYAL2	Forward	5’-TTCTACCGCGACCGTCTAGG-3’
	Reverse	5’-TGTCCGAATGTAGTGCTCCAC-3’
HYAL3	Forward	5’-TGCCTCTTCTTTTCCCTGCTG-3
	Reverse	5’-GGGAGGGTTGACTGTAAACTG-3
RPLPO	Forward	5’- TTCGACAATGGCAGCATCTACAA -3’
	Reverse	5’- CTGCAGACAGACACTGGCAACA -3’

## Data Availability

All data are contained within the manuscript.

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
