# Peer review of "The Attenuated Secretion of Hyaluronan by UVA-Exposed Human Fibroblasts Is Associated with Up- and Downregulation of HYBID and HAS2 Expression via Activated and Inactivated Signaling of the p38/ATF2 and JAK2/STAT3 Cascades"

_ijms, 2021, doi:10.3390/ijms22042057_

Round 1
Reviewer 1 Report
This is a very well written and interesting manuscript. Methods and results are clarly showed.
Author Response
Thank you for your comments.
Reviewer 2 Report
- Several western blot results were represented with combined with two separated bands such as Figure 8H. This could be critical issue. And we cannot get any conclusion with separated experiments.
Author Response
Thank you for your comments.
Considering the reviewer's comments, we deleted Western blotting image for 15 min post-UVA irradiation in Fig7H (not Fig8H) in the results section as marked by red in the revised manuscript with tracking changes.
Reviewer 3 Report
I am very happy with the subject that has been exposed. interesting and well writtenAuthor Response
Thank you for your comments.
Reviewer 4 Report
The importance of the research and the research method are okay. However, the data processing method, data processing method, and thesis writing are very rudimentary. Authors should rewrite this paper by referring to several other papers in IJMS.
Author Response
Reviewer #4
The importance of the research and the research method are okay. However, the data processing method, data processing method, and thesis writing are very rudimentary. Authors should rewrite this paper by referring to several other papers in IJMS.
Response:
We are sorry to mention that we believe that our manuscript has no problem with data processing method and thesis writing. However, considering the reviewer’s comments, we have extensively modified western blotting images as marked by red in the re-revised manuscript with remarks.
Round 2
Reviewer 4 Report
The authors ignored the reviewer's proposal and presented it almost as it was. I would like to see a paper recently published by IJMS and change it to a similar format. I would like to reduce the figure to 5 lines and change the way of showing the data to a general way. I don't think I'm looking at a lab note, not a paper.
Authors should be read recent paper published in IJMS
https://www.mdpi.com/search?q=&journal=ijms&sort=pubdate&page_count=50
Author Response
Response:
According to the reviewer 4, we have changed our manuscript to IJMS format as marked by tracking changes in the R3 revised manuscript.
